

# Comprehensive study of serine/arginine-rich (SR) gene family in rice: characterization, evolution and expression analysis

Rui Gao, Yingying Lu, Nan Wu, Hui Liu and Xiaoli Jin

Department of Agronomy, The Key Laboratory of Crop Germplasm Resource of Zhejiang Province, Zhejiang University, Hangzhou, China

## ABSTRACT

As important regulators of alternative splicing (AS) events, serine/arginine (SR)-rich proteins play indispensable roles in the growth and development of organisms. Until now, the study of SR genes has been lacking in plants. In the current study, we performed genome-wide analysis on the SR gene family in rice. A total of 24 *OsSR* genes were phylogenetically classified into seven groups, corresponding to seven subfamilies. The *OsSR* genes' structures, distribution of conserved domains, and protein tertiary structure of OsSR were conserved within each subfamily. The synteny analysis revealed that segmental duplication events were critical for the expansion of *OsSR* gene family. Moreover, interspecific synteny revealed the distribution of orthologous SR gene pairs between rice and *Arabidopsis*, sorghum, wheat, and maize. Among all *OsSR* genes, 14 genes exhibited NAGNAG acceptors, and only four *OsSR* genes had AS events on the NAGNAG acceptors. Furthermore, the distinct tissue-specific expression patterns of *OsSR* genes showed that these genes may function in different developmental stages in rice. The AS patterns on the same *OsSR* gene were variable among the root, stem, leaf, and grains at different filling stages, and some isoforms could only be detected in one or a few of tested tissues. Meanwhile, our results showed that the expression of some *OsSR* genes changed dramatically under ABA, GA, salt, drought, cold or heat treatment, which were related to the wide distribution of corresponding *cis*-elements in their promoter regions, suggesting their specific roles in stress and hormone response. This research facilitates our understanding of SR gene family in rice and provides clues for further exploration of the function of *OsSR* genes.

## INTRODUCTION

Splicing of pre-mRNA is an important post-transcriptional regulatory mechanism in eukaryotes, a pre-mRNA can produce different transcripts by splicing at different splicing sites. It was reported that more than 40% genes in plants undergo AS (*Chen et al., 2019a*). After transcription, the splicing of pre-mRNA is crucial to the production of mature mRNA, which takes place in the spliceosome. The core of the spliceosome was composed

Corresponding author
Xiaoli Jin, jinxl@zju.edu.cn

of five small nuclear ribonucleoproteins (U1, U2, U4, U5, and U6 snRNPs) and numerous non-snRNP. SR proteins were important non-snRNP that regulate splicing of pre-mRNA (*Long & Caceres, 2009*; *Wang & Brendel, 2004*). The serine/arginine (SR) proteins are well-known splicing factors that play important roles in both the assembly of spliceosomes and the regulation of alternative splicing (AS) (*Long & Caceres, 2009*).

The serine/arginine (SR) proteins were characterized by the presence of RNA recognition motif (RRM) and the consecutive serine and arginine dipeptides repeats and RNA recognition motif (RRM) (*Barta, Kalyna & Reddy, 2010*). The consecutive serine and arginine dipeptides repeat functioned as a protein-interaction domain and the RRM provided RNA-binding specificity for SR proteins. Besides, some SR proteins had specific domains which were less well understood, like Zn-knuckles and RGG box (*Jin, 2022*). The results of plant SR proteins sequence comparison and phylogenetic analysis showed that plant SR proteins were classified into seven different subfamilies, SR, SR45, RSZ, SC, SCL, RS2Z and RS, and the last three of these subfamilies were plant-specific which had unique domains, while the other four subfamilies had orthologs in animals (*Reddy & Shad Ali, 2011*; *Richardson et al., 2011*). SR proteins in SCL subfamily had an RRM with a charged extension at the N-terminus. The members of SCL subfamily included dicotyledons, monocotyledons, mosses and green algae. The RS2Z subfamily was found in dicotyledons and monocotyledons, and two Zn-knuckles domains and an extra SP-rich region were present on the proteins in this subfamily. The members of RS subfamily in plants contained two RRMs and the RS domains which were rich in RS dipeptides. The RS subfamily was mainly composed of photosynthetic eukaryotes (*Xie et al., 2022*).

The SR gene itself undergoes extensive AS. It was shown that 18 SR genes in *Arabidopsis thaliana* could produce more than 90 transcripts, and the precursor mRNA from the rice SR gene also underwent extensive AS (*Reddy & Shad Ali, 2011*). Alternative splicing (AS) events that occurred at the NAGNAG acceptor were termed the AS-NAGNAG events, which would cause NAG insertion-deletions in transcripts (*Iida, Shionyu & Suso, 2008*). In the NAGNAG motif, the first AG is termed the E-acceptor and the second AG was termed I-acceptor (*Hiller et al., 2004*). AS-NAGNAG events were widespread in mammals and plants, which contributes to the diversity of transcriptome and proteome in different species. It has been reported that AS-NAGNAG acceptors were overrepresented in genes which coded RRM-containing proteins. Genes coding for RNA binding proteins were preferentially equipped with NAGNAG acceptors in human (*Akerman & Mandel-Gutfreund, 2006*; *Iida, Shionyu & Suso, 2008*). In *Arabidopsis thaliana*, NAGNAG acceptors were frequently found in the genome, particularly in the SR genes (*Schindler et al., 2008*; *Shi, Sha & Sun, 2014*).

Because of the indispensable role played by SR proteins in both constitutive splicing and alternative splicing of precursor mRNA, it is likely that they will be instrumental in modulating the expression of genes that are essential for plants in different developmental stages. Until now, some studies have revealed that members of the SR gene family played important roles in various biological processes in different species (*Ali et al., 2007*; *Carvalho, Carvalho & Duque, 2010*; *Park et al., 2020*). In *Arabidopsis*, 19 *AtSR* genes have been identified (*Barta, Kalyna & Reddy, 2010*). The function of *SR45* has been extensively studied in *Arabidopsis thaliana*. Studies have shown that *SR45* negatively regulated glucose-induced
growth by inhibiting abscisic acid (ABA) accumulation and its signaling, thereby inhibiting seedling establishment under adverse conditions (*Ali et al., 2007*; *Carvalho, Carvalho & Duque, 2010*). *AtSR45a* was detected to undergo AS and produced two alternative splicing variants, *AtSR45a-1a* and *AtSR45a-1b*. *AtSR45a* could regulate the response to salt stress by increasing the expression of these two variants and interacting with the cap-binding complex (*Li et al., 2021*). *AtRS40*, *AtRS41*, and *AtSCL30* participated in responses to ABA and salt stress in *Arabidopsis* (*Chen et al., 2013*; *Cruz et al., 2014*). Besides, *AtSR* genes including *AtRS40*, *AtSR34a*, *AtRSZ22*, and *AtSR45a etc.*, were reported to participate in the response to heat stress. Under high temperature, these genes underwent specific AS and produced specific mRNA variants (*Filichkin et al., 2010*; *Ling, Mahfouz & Zhou, 2021*; *Ling et al., 2018*).

In rice, 24 SR genes have been identified (*Barta, Kalyna & Reddy, 2010*). The *OsRSp29*, *OsRSZp23*, and *OsSCL26* played roles in stimulating pre-mRNA splicing and promoting the splicing efficiency of downstream genes (*Isshiki, Tsumoto & Shimamoto, 2006*). The OsSR45 functioned in regulating the response to various stresses, including temperature stress and reactive oxygen species stress at the post transcriptional level by interacting with OsFKBP20-1b which belonged to the immunophilin family in rice (*Park et al., 2020*). *OsSR40*, *OsSCL57*, and *OsSCL25* played crucial roles in regulating mineral element absorption and homeostasis in rice by participating in alternative splicing of pre-mRNAs of related genes (*Dong et al., 2018*).

Despite the fact that SR genes have been identified in rice, further studies on these genes, especially on their biological functions, are still lacking. A comprehensive analysis of SR genes in rice was performed in the present study. The genetic relationship among *OsSR* genes was analyzed firstly, then we analyzed the structures of the OsSR proteins, the collinear relationships, as well as the promoter sequences, the NAGNAG acceptors, and the expression and alternative splicing patterns in both vegetative and reproductive organs of *OsSR* genes, and their responses to hormones and abiotic stresses. This study provides a better understanding and establishes the foundation for further functional elucidation of *OsSR* genes.

## MATERIALS AND METHODS

### Identification and acquisition of information of *OsSR* genes

According to the accession number provided in Jin's article (*Jin, 2022*), we extracted the sequences of 24 *OsSR* genes, transcript sequences produced by each gene, and the corresponding encoded protein sequences from the MSU-RGAP (Rice Genome Annotation Project, RGAP) database (http://rice.uga.edu/index.shtml). Meanwhile, the accession numbers of the corresponding genes on the Rice Genome Annotation Project Database (https://rapdb.dna.affrc.go.jp/) were also provided in this study (*Kawahara et al., 2013*; *Sakai et al., 2013*). We summarized the alternative splicing information of 24 *OsSR* genes based on the two databases, MSU-RGAP and RAP DB, which were shown in Table S1. The AS models of *OsSR* genes except *OsRS2Z39* were different in the MSU-RGAP database and RAP DB. We found that the splicing forms of SR genes from the MSU-RGAP database

comprised more splicing isoforms compared to the RAP DB, and the alternative splicing information of these genes in the RAP DB was almost included in MSU-RGAP. Moreover, we analyzed the cDNA information in the NBCI datebase with the two databases, which showed identity in MSU-RGAP. Thus, we investigated the alternative splicing pattern as well as the expression of different transcripts produced by the *OsSR* genes based on the AS models of all 24 *OsSR* genes on the MUS-RGAP.

## Bioinformatics analysis of SR gene family genes
### Phylogenetic analysis, conserved motifs, gene structure, tertiary structure prediction

Based on the results of the multiple amino acid sequence alignment done by MEGA 7.0 with ClustalW, MEGA 7.0 was used in this study to perform the phylogenetic analysis with the maximum-likelihood (ML) method and 1,000 bootstrap replicates (*Kumar, Stecher & Tamura, 2016*). Exon and intron positions in *OsSR* genes were mapped, and gene structures were deduced using the Gene Structure Display Server (GSDS) (http://gsds.cbi.pku.edu.cn/) (*Hu et al., 2015*). Conserved motifs of OsSR proteins were analyzed using SMART (Simple Modular Architecture Research Tool) online tool (http://meme-suite.org/tools/meme) (*Letunic & Bork, 2018*), and then visualized using the TBtools (*Chen et al., 2020a*). The tertiary structures of OsSR proteins were predicted by SWISS-MODEL (https://swissmodel.expasy.org/) (*Waterhouse et al., 2018*).

### Physicochemical properties and subcellular localizations

The physicochemical properties and subcellular localizations of OsSR proteins were predicted using ExPASy Protparam online (https://web.expasy.org/protparam/) and BUSCA (http://busca.biocomp.unibo.it) (*Savojardo et al., 2018*), respectively. The NetPhos3.1 service was used to predict the OsSR proteins' phosphorylation sites (https://services.healthtech.dtu.dk/service.php?NetPhos-3.1) (*Blom, Gammeltoft & Brunak, 1999*). Meanwhile, the subcellular localization of OsSR proteins was analyzed online (https://croppal.org/) (*Hooper et al., 2020*).

### Syntenic relationships

From the Ensembl plants database, the genomic information of SR genes in rice, *Arabidopsis*, sorghum, maize, soybean, and wheat was retrieved (http://plants.ensembl.org/index.html). Then, the segment duplication events of *OsSR* genes and synteny relationships between rice and other species were analyzed by the MCScanX program, and the results were visualized using TBtools (*Chen et al., 2020a*; *Wang et al., 2012*).

### Cis-acting elements

The key promoter regions (2,000 bp sequences upstream of the translation start codons) of the *OsSR* genes were retrieved from NCBI (https://www.ncbi.nlm.nih.gov/gene/) and submitted to the PlantCARE online (http://bioinformatics.psb.ugent.be/webtools/plantcare/html/) to predict and analyze the regulatory *cis*-acting elements in promoter regions (*Lescot et al., 2002*).

## Plant materials and growth conditions

Rice (*Oryza sativa* L. spp. Japonica, var Nipponbare) plants were used in this study. The seeds were sterilized with 20% NaClO solution, soaked with sterile water and incubated at 37 °C for germination. A portion of seeds were transferred to the field with normal water and fertilizer management to continue growing until maturity. Meanwhile, the other portion was transplanted to the 96-well PCR plates with the bottom removed, and grown in the Hoagland solution ($CaNO_3 \cdot 4H_2O$ 945 mg/L, $KNO_3$ 506 mg/L, $NH_4NO$ 380 mg/L, $KH_2PO_4$ 136 mg/L, $MgSO_4 \cdot 7H_2O$ 493 mg/L, iron salt solution 2.5 mL/L (2.78 g $FeSO_4 \cdot 7H_2O$, 500 mL distilled water, 3.73 g EDTA-2Na pH 5.5), microelement five mL/L (KI 0.83 mg/L, $H_3BO_3$ 6.2 mg/L, $MnSO_4$ 22.3 mg/L, $ZnSO_4$ 8.6 mg/L, $Na_2MoO_4$ 0.25 mg/L, $CuSO_4$ 0.025 mg/L, $CoCl_2$ 0.025 mg/L), pH 6.0) in the greenhouse with a photoperiod of 14/10 h at 28 °C/26 °C (day/night) and relative humidity of 65%. The Hoagland solution was renewed every 2 days.

For tissue-specific expression analysis, the different tissues of rice planted in the field were sampled from vegetative organs and spikelets at different filling stages. For the salt, drought, and phytohormone treatments, the seedlings at the 4-leaf stage were subjected to Hoagland solution with 200 mM NaCl, 20% PEG6000, 100 μM gibberellin (GA) (100 μM) or abscisic acid (ABA), respectively. For cold or heat stress, the seedlings at the 4-leaf stage were moved to the incubator with the temperature of 4 °C and 37 °C, respectively. And the 2nd and 3rd leaves from the seedlings were dissected at different time points under various treatments. The samples without treatments were set with blank control (CK) at the same time. In the CK treatment, the seedlings at the 4-leaf stage were subjected to Hoagland solution and grew in the same incubator as the corresponding treated group.

## RNA isolation, RT-PCR and *q*RT-PCR

Total RNA from different samples was extracted by Total RNA Extractor (Trizol) (Sangon Biotech, Shanghai, China) and reverse transcribed into cDNA using Hifair® III; Reverse Transcriptase (Yeasen, Shanghai, China) according to the instruction book.

The RT-PCR was performed using PrimerSTAR MaxDNA Polymerase (TaKaRa, Shiga, Japan), and the appropriate annealing temperature for PCR according to the properties of primer pairs for different genes. The number of amplification reaction cycles in this study was 30. The RT-PCR experiment for each gene was conducted with 3 biological replicates. The PCR products were determined using electrophoresis on the 1% agarose gels. Specific primers for different *OsSR* genes and the control gene *OsActin* used for RT-PCR are listed in Table S7.

For the *q*RT-PCR experiment, SYBR Green qPCR Master Mix (TOROIVD) was used, and the experiment was conducted on LightCycler 480 II (Roche, South San Francisco, CA, USA). The data was analyzed as previously described using *OsActin* as the internal standard (*Livak & Schmittgen, 2001*). The *q*RT-PCR experiment was carried out using 3 biological replicates, and 3 technical replicates were performed for each biological replicate. The *q*RT-PCR data was calculated using $2^{-\Delta\Delta CT}$ method and Student's *t*-test. The primer sequences for *q*RT-PCR are listed in Table S6.
## RESULTS

### Identification of *OsSR* genes and their characteristics

Here, we performed the prediction and analysis of the physicochemical properties of the OsSR proteins (Table 1). The results revealed that OsSR proteins ranged in length from 185 amino acids (OsRSZ21a and OsRSZ21) to 502 amino acids (OsSCL57), while the molecular weight varied from 21.02 kDa (OsRSZ21) to 56.83 kDa (OsSCL57). Notably, all of the rice SR proteins were alkaline proteins with the isoelectric point ranging from 8.67 (OsSR40) to 12.37 (OsSR45) (Table 1).

According to the BUSCA prediction analysis, OsSR45 was predicted to localize in the chloroplast and nucleus, the remaining 23 OsSR proteins were localized only in nucleus. By the experiments, OsSR45 was observed to co-express and physically interact with OsFKBP20-1b in the nucleus and cytoplasm *in vivo* (*Park et al., 2020*). It has been proven experimentally that OsSCL30, which was the SR protein in rice, was visible only in the nucleus (*Zhang et al., 2022*). The results indicated that different OsSR proteins together with their interaction proteins could function in different intracellular partitions. Besides, the CropPAL data set showed that the subcellular localization of some OsSR proteins had been confirmed by mass spectrometric assay, including OsSCL30a, OsRS2Z37, OsRSZ21, OsRSZ23, OsRS33, and OsSR33 (Table 1).

According to previous research, the mobility of SR proteins was regulated by phosphorylation in *Arabidopsis* (*Tillemans et al., 2006*). Notably, we found that all of the OsSR proteins were phosphorylated with numbers of phosphorylation sites ranging considerably from 25 (OsSC25) to 86 (OsSR45a) (Table 1). The difference in physicochemical properties is suggestive of functional differences among the OsSR proteins.

### Phylogenetic, motif composition, structure of OsSR proteins and gene structure analysis of *OsSR* genes

Analysis of the evolutionary relationships among the members of the SR family in rice by a maximum-likelihood phylogenetic tree using amino acid sequences of 24 OsSR proteins was displayed (Fig. 1). A total of 24 *OsSR* genes could be classified into seven distinct subgroups based on the evolutionary relationships, and the results were consistent with the reported seven subfamilies: SCL, SC, SR45, RS2Z, RSZ, RS, and SR, indicating their conservation within the subfamilies during their evolution. These 7 subgroups contained 6, 3, 2, 4, 3, 2, and four members, respectively. Among them, SR, RSZ, and SC subfamily are common between plants and animals, the remaining subfamilies are plant-specific (*Reddy & Shad Ali, 2011*), indicating the SR proteins diverged along with the speciation.

The conserved domains usually have important functions and are closely related to the completion of physiological functions of proteins. Analysis of the conserved domains of OsSR proteins showed that the OsSR proteins within the same subfamily were highly conserved in the types and distribution of conserved domains (Fig. 1B). The members of the SCL and SC subfamilies contained only one RRM domain near the N-terminus, as well as a domain rich in consecutive serine and arginine dipeptide repeats (SR domain) at the C-terminus of the protein (*Jin, 2022*). Besides, the OsSR proteins in the SCL subfamily

Gao et al. (2023), PeerJ, DOI 10.7717/peerj.16193

**Table 1** Basic information of *OsSR* gene family members.

| Subfamily | Revised nomenclature | RAP_Locus | MSU_Locus (Alternative splice form) | Length (bp) | Intron | Exon | Protein length (aa) | Molecular weight (kDa) | Isoelectric point | Instability index | Predicted subcellular location | phosphorylation sites |
|---|---|---|---|---|---|---|---|---|---|---|---|---|
| | *OsSCL30a* | Os02g0252100 | LOC_Os02g15310.1 | 4,309 | 6 | 7 | 265 | 30.47 | 11.09 | 111.87 | *plasma membrane\*; nucleus\** | 45 |
| | *OsSCL30* | Os12g0572400 | LOC_Os12g38430.1 | 3,856 | 6 | 7 | 263 | 30.19 | 10.9 | 110.93 | ***nucleus*** | 47 |
| | *OsSCL57* | Os11g0704700 | LOC_Os11g47830.1 | 7,034 | 13 | 14 | 502 | 56.83 | 10.01 | 100.02 | nucleus | 82 |
| SCL | *OsSCL28* | Os03g0363800 | LOC_Os03g24890.1 | 4,646 | 5 | 6 | 243 | 27.78 | 10.83 | 88.03 | nucleus | 35 |
| | *OsSCL25* | Os07g0633200 | LOC_Os07g43950.1 | 3,180 | 4 | 5 | 213 | 24.82 | 10.68 | 103.01 | nucleus | 50 |
| | *OsSCL26* | Os03g0374575 | LOC_Os03g25770.1 | 3,549 | 4 | 5 | 218 | 25.68 | 11.17 | 114.71 | nucleus | 40 |
| SC | *OsSC32* | Os07g0623300 | LOC_Os07g43050.1 | 4,182 | 7 | 8 | 275 | 32.24 | 11.35 | 118.57 | nucleus | 59 |
| | *OsSC34* | Os08g0486200 | LOC_Os08g37960.1 | 3,154 | 6 | 7 | 289 | 33.54 | 11.8 | 112.71 | nucleus | 56 |
| | *OsSC25* | Os03g0388000 | LOC_Os03g27030.1 | 3,000 | 6 | 7 | 206 | 24.90 | 10.33 | 64.44 | nucleus | 25 |
| SR45 | *OsSR45a* | Os05g0105900 | LOC_Os05g01540.1 | 3,826 | 10 | 11 | 426 | 47.75 | 12.19 | 133.86 | nucleus | 86 |
| | *OsSR45* | Os01g0959000 | LOC_Os01g72890.1 | 5,025 | 11 | 12 | 432 | 48.11 | 12.37 | 160.81 | ***nucleus*, *cytoplasm*** | 77 |
| RS2Z | *OsRS2Z39* | Os05g0162600 | LOC_Os05g07000.1 | 4,237 | 5 | 6 | 338 | 39.02 | 9.83 | 64.03 | nucleus | 45 |
| | *OsRS2Z37* | Os01g0155600 | LOC_Os01g06290.1 | 3,944 | 5 | 6 | 324 | 36.89 | 11.27 | 96.97 | *nucleus\** | 57 |
| | *OsRS2Z36* | Os05g0120100 | LOC_Os05g02880.1 | 3,489 | 5 | 6 | 323 | 36.22 | 10.83 | 101.82 | nucleus | 61 |
| | *OsRS2Z38* | Os03g0285900 | LOC_Os03g17710.1 | 3,333 | 5 | 6 | 335 | 37.52 | 11 | 100.05 | nucleus | 60 |
| RSZ | *OsRSZ21a* | Os06g0187900 | LOC_Os06g08840.1 | 3,671 | 4 | 5 | 185 | 21.18 | 11.29 | 103.17 | nucleus | 34 |
| | *OsRSZ21* | Os02g0789400 | LOC_Os02g54770.1 | 4,678 | 4 | 5 | 185 | 21.02 | 11.24 | 93.82 | *nucleus \** | 29 |
| | *OsRSZ23* | Os02g0610600 | LOC_Os02g39720.2 | 4,269 | 3 | 4 | 203 | 23.20 | 11.33 | 112.07 | *plastid\*; nucleus\** | 35 |
| RS | *OsRS29* | Os04g0118900 | LOC_Os04g02870.1 | 4,490 | 4 | 5 | 245 | 28.78 | 9.94 | 68.28 | nucleus | 31 |
| | *OsRS33* | Os02g0122800 | LOC_Os02g03040.1 | 3,691 | 4 | 5 | 279 | 32.54 | 9.88 | 60.31 | *nucleus\** | 30 |
| SR | *OsSR33* | Os07g0673500 | LOC_Os07g47630.1 | 5,111 | 12 | 13 | 296 | 33.14 | 10.64 | 104.32 | *nucleus \** | 59 |
| | *OsSR32* | Os03g0344100 | LOC_Os03g22380.1 | 4,789 | 12 | 13 | 286 | 31.94 | 10.54 | 98.23 | nucleus | 55 |
| | *OsSR33a* | Os05g0364600 | LOC_Os05g30140.1 | 7,169 | 13 | 14 | 294 | 33.42 | 10.92 | 102.83 | nucleus | 64 |
| | *OsSR40* | Os01g0316550 | LOC_Os01g21420.1 | 7,570 | 12 | 13 | 292 | 33.50 | 8.67 | 48.14 | nucleus | 29 |

**Notes.**

(1) The subcellular location was bolded and italicized the corresponding OsSR protein has been experimentally verified.

(2) The subcellular location of the OsSR proteins, determined by mass spectrometric assay, was italicized and marked with *.

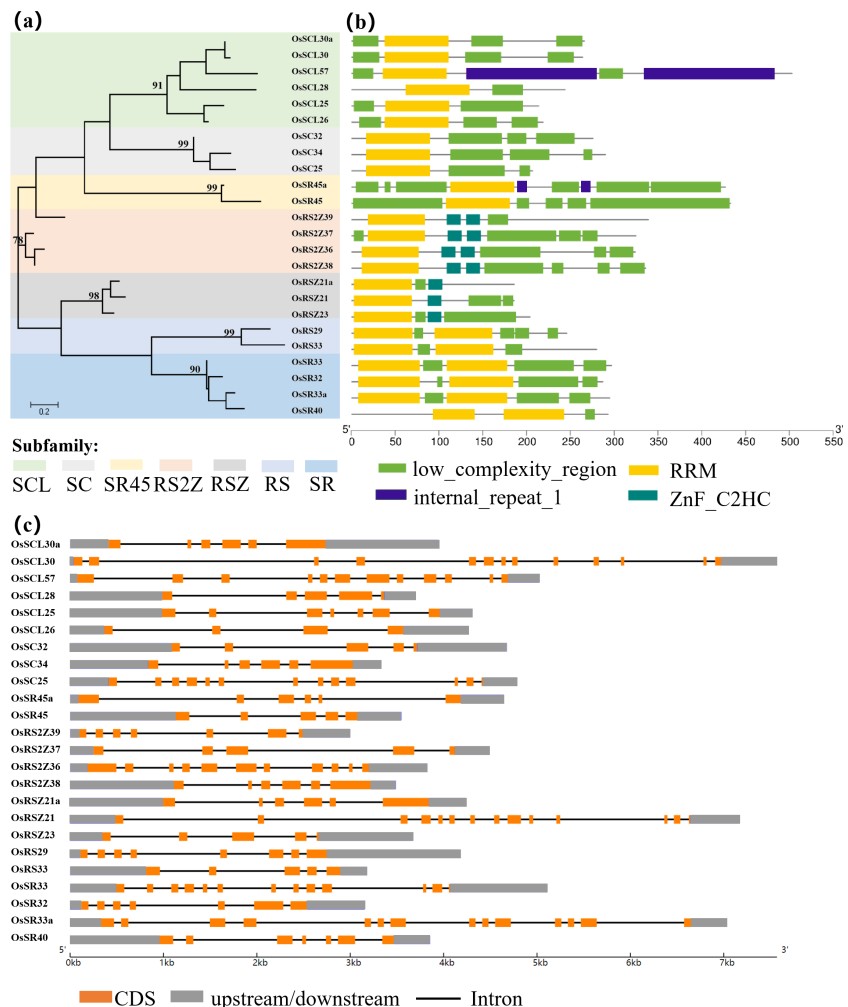

**Figure 1** **Schematic representation of the phylogenetic relationship, gene structures and conserved motifs in *OsSR* genes.** (A) Maximum-likelihood (ML) phylogenetic tree of OsSR proteins. (B) Distribution of conserved motifs in OsSR proteins. (C) Exon/introns and untranslated regions (UTRs) of *OsSR* genes. Grey boxes denote UTR (untranslated region); yellow boxes denote exon; black lines denote introns. The length of gene can be estimated using the scale at the bottom.

also had a domain which could be diverse at the N-terminal. As for two members of the SR45 subfamily, both of them contained one RRM domain and two SR domains which existed in the N- and C-termini of the proteins, respectively. Unlike OsSR proteins in other subfamilies, two and one ZnF_C2HC motif were contained in members of the RS2Z and RSZ subfamily besides RRM and SR domains, respectively. The members in RS subfamily contained two RRMs and the RS domain. Similarly, the four proteins of the SR subfamily also contained two RRM domains, but unlike the RS subfamily, the C-terminal of these proteins was the SR domain.

Furthermore, the prediction of the tertiary structure of these proteins using SWISS-MODEL online server revealed that OsSR proteins were mainly composed of α-helices,

β-folds, and random coils (Fig. S1 and Table S2). It was speculated that proteins with different tertiary structures may determine the diversity functions of *OsSR* genes. We noticed that OsSR protein structures showed differences among different subfamilies, especially among RS subfamily, SR subfamily and other subfamilies (Fig. S1). While in most cases, the OsSR proteins in the same subfamily had similar tertiary structures (Fig. S1).

Gene structural variety might function as a form of evolution for numerous genes (*Fedorov, Merican & Gilbert, 2002*). The conservation of gene structure is related to the number of introns in eukaryotes (*Rogozin et al., 2003*). Further exploration of the gene revealed the structural differences and conserved relationships among these *OsSR* genes. The *OsSR* genes differed in nucleotide sequence, but they contained the similar number of exons and introns in the same subfamily except for the genes in the SCL subfamily. The number of introns among different subfamilies ranged from 3 to 13 (Fig. 1C). The genes, containing the fewest introns and exons and the most introns and exons, were *OsRSZ23* and *OsSR33a*, belonging to RSZ and the SR subfamily, respectively. The *OsSR* genes in the SC subfamily usually contained six or seven introns, while the members of the SR45 subfamily contained 10 or 11 introns. All the members of RS2Z and RS subfamily contained five and four introns, respectively. The number of introns of *OsSR* genes in the RSZ subfamily is four or three. The intron number in SR subfamily was up to 12 or 13.

## Segment duplication analysis of *OsSR* genes

In order to clarify the expansion of rice SR gene family, we analyzed segmental duplication, which is considered as one of the main factors driving the expansion of gene families during evolution in plants (*Cannon et al., 2004*). As shown in Fig. 2, 12 genes (six pairs), including *OsRS2Z37* and *OsRS2Z39*, *OsSR40* and *OsSR33a*, *OsSR45* and *OsSR45a*, *OsRSZ21* and *OsRSZ21a*, *OsSR32* and *OsSR33*, *OsSCL57* and *OsSCL30*, are implicated in segmental duplication events. Totally, 50% members of *OsSR* genes showed collinear relationships, indicating that segmental duplication was primarily responsible for the expansions of the SR gene family in rice. Additionally, we found that the *Ka/Ks* ratios of all *OsSR* gene duplication pairs were less than 1.0 (Table S3), suggesting that these six duplication pairs underwent purifying selection (*Hurst, 2002*).

## Synteny and orthologous gene pairs of SR genes

The syntenic relationship of the SR genes between rice and five plant species (*Sorghum bicolor*, *Arabidopsis thaliana*, *Zea mays*, *Triticum aestivum*, and *Glycine max*) was examined in this study. There was no orthologous gene between rice and soybean (Fig. S2). While only four pairs of orthologous genes were identified between rice and *Arabidopsis* (*OsSR33a* and *AtSR34*, *OsSR33* and *AtSR34*, *OsRS2Z36* and *AtRS2Z33*, *OsRS2Z36* and *AtRS2Z32*) (Fig. 3). In addition, a total of 20, 40, 64 orthologous SR gene pairs were identified between rice and sorghum, maize, wheat, respectively (Fig. 3 and Table S4).

## NAGNAG acceptors in *OsSR* genes

NAGNAG acceptors were termed based on the existence of a NAGNAG acceptor motif, and alternative splicing at NAGNAG acceptors was widespread in the genomes of animals and plants (*Akerman & Mandel-Gutfreund, 2006*). A scan of *OsSR* gene products from

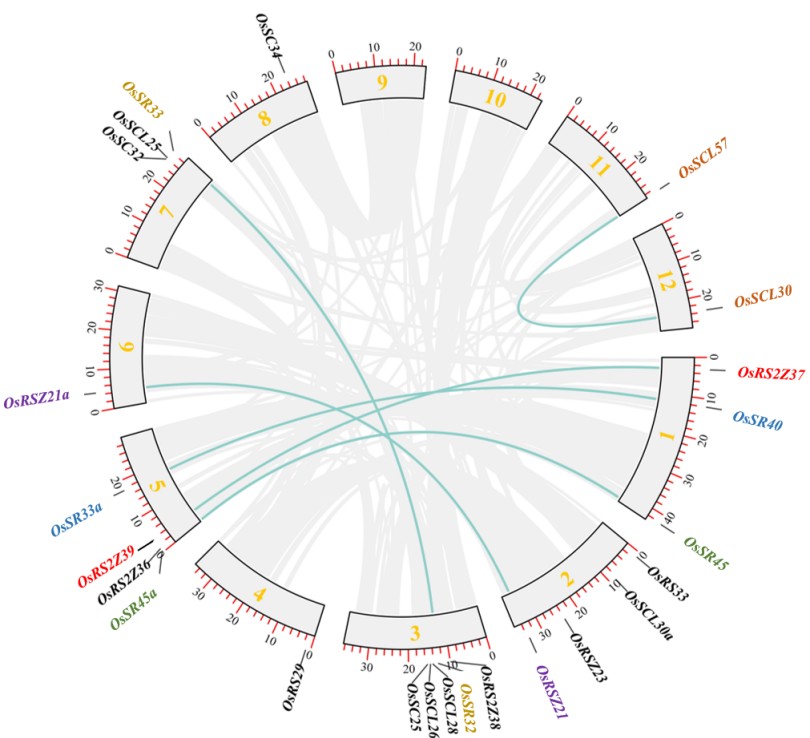

**Figure 2  Schematic representations for the chromosomal locations and segment duplications of *OsSR* genes.** A total of 24 *OsSR* genes were mapped onto the chromosomes on the basis of their physical location. 1–12 were the chromosome numbers (Chr1- Chr12). The gray lines indicated duplicated blocks. The duplicated *OsSR* gene pairs were highlighted in green lines.

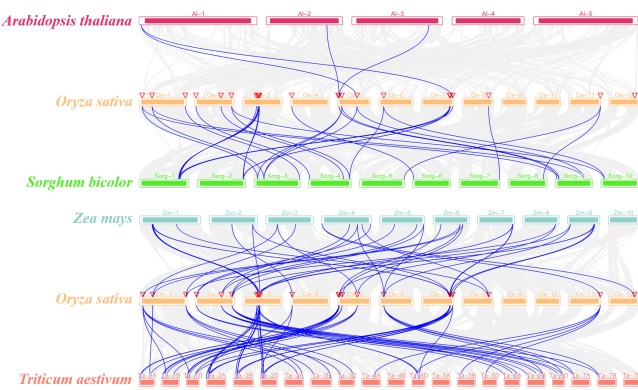

**Figure 3  Synteny analyses of SR genes between rice and four plant species (*Arabidopsis thaliana*, *Sorghum bicolor*, *Zea mays*, and *Triticum aestivum*).** The gray lines indicated collinear blocks and syntenic SR gene pairs would be highlighted in blue lines.

**Table 2   NAGNAG acceptors in *OsSR* genes.**

| SR gene | mRNA | | Motif |
|---------|---|---|-------|
|         | E | I |       |
| *OsSCL30a* | 1 | | CAGGAG |
| *OsSCL30* | 1 | | CAGGAG |
| *OsSCL26* | 2 | | CAGTAG, TAGCAG |
| *OsSC25* | 1 | | TAGCAG |
| *OsSR45a* | 1 | | CAGCAG |
| *OsSR45* | 1 | | CAGCAG |
| *OsRS2Z39* | 1 | | CAGCAG |
| *OsRS2Z36* | 2 | | CAGGAG |
| *OsRS2Z38* | 1 | | CAGGAG |
| *OsRSZ21a* | 1 | 2 | CAGAAG, GAGCAG, AAGCAG |
| *OsRSZ21* | 2 | | CAGAAG |
| *OsRSZ23* | 1 | | TAGGAG |
| *OsRS33* | 1 | | CAGGAG |
| *OsSR33* | 1 | | CAGAAG |

**Notes.**
   Observed NAGNAG motifs and E and I acceptors confirmed by mRNA (from RefSeq) are shown.

the information on the MSU-RGAP for signatures associated with NAGNAG acceptors revealed that 14 out of 24 *OsSR* genes exhibited NAGNAG acceptors. Among these 14 genes, *OsSCL26*, *OsRS2Z36*, *OsRSZ21a* and *OsRSZ21* contained 2, 2, 3, 2 NAGNAG acceptors, respectively, and the other 10 genes contained only one (Table 2). Furthermore, we focused on whether alternative splicing occurred on these NAGNAG acceptors. We found that only four *OsSR* genes, including *OsSCL26*, *OsSC25*, *OsSR45a* and *OsSR45*, had AS-NAGNAG event on the NAGNAG acceptor, which led to the deletion of a single amino acid at the protein level (Table S1).

## The prediction of *cis*-acting elements on *OsSR* genes' promoters

The analysis of the promoter regions of *OsSR* genes identified six types of *cis*-elements, including promoter/enhancer elements and elements related to light responsiveness, stress, hormone response, development/tissue specificity or circadian control (Fig. 4 and Table S5). The proportion of *cis*-acting elements in each of these categories was 9.1%, 45.5%, 9.1%, 18.2%, 16.4%, and 1.8%, respectively.

   The *cis*-acting elements in 'promoter/enhancer element' category, which ensure the correct location and start of transcription, were ubiquitously identified in all *OsSR* genes' promoters, including CAAT-box, and TATA-box (Fig. 4). The ARE and MBS in 'stress' category, which are involved in anaerobic induction and drought responsiveness, respectively, were harbored in most of *OsSR* genes' promoters. In 'hormone response' category, the *cis*-acting elements that respond to ABA, auxin, GA, MeJA, and salicylic acid were identified. The ABA, salicylic acid and MeJA responsiveness elements, including the ABA responsive element (ABRE), TCA-motif, TGACG-motif, and CGTCA-motif, were widely presented in the promoters of the *OsSR* genes. Notably, the ABRE was the

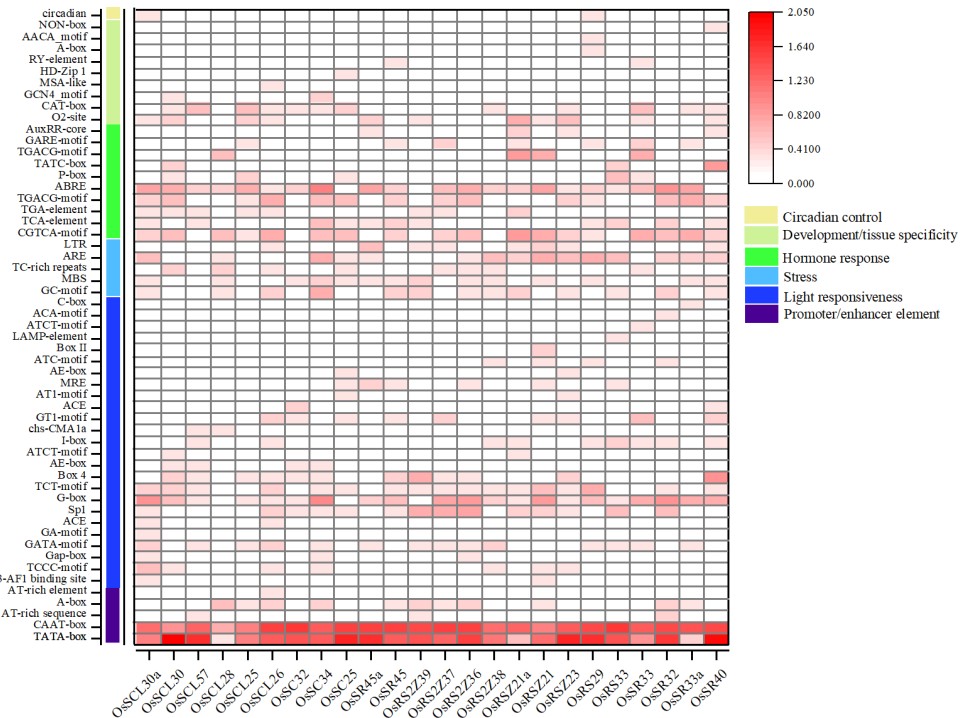

**Figure 4** *Cis-acting elements in promoter regions of OsSR genes.* *Cis*-acting elements were predicted based on 2 kb sequences upstream of coding sequences. The quantity of *cis*-acting elements was normalized by log 10 (number + 1) and then used for heatmap construction.

most widely distributed hormone-responsive element, which presented in almost all the promoters of the 24 *OsSR* genes. Among these hormone-responsive elements, GA responsive elements were the most abundant. We found that three out of nine identified hormone-responsive elements were GA response elements (Fig. 4 and Table S5), including P-box, TATC-box, and GARE-motif. As for the 'development/tissue specificity' category, GCN4_motif, RY-element, and AACA_motif were identified as seed and endosperm development-related (Fig. 4 and Table S5). In addition, only *OsSCL30a* and *OsRS29* contained circadian related elements in their promoters. Based on these findings, it indicated that *OsSR* genes may have roles in responding to different hormones and environmental stresses.

## Expression patterns and AS of *OsSR* genes

A series of *cis*-acting elements related to tissue development, stress response, and hormone response were identified in the promoter region of the *OsSR* genes. To understand whether the *OsSR* genes are implicated in the growth and abiotic stress response of rice, we examined the expression profiles and AS patterns of *OsSR* genes in several tissues or under abiotic conditions (drought, salt, cold, and heat) and hormone treatments (GA and ABA).

### Tissue expression profiles of the OsSR genes

To further characterize the potential biological function, *q*RT-PCR was used to conduct tissue-specific expression analyses of *OsSR* genes. Totally, we detected all 24 *OsSR* genes in eight tissues and organs including root, stem, leaf and spikelets before fertilization, at flowering and 5, 10 and 20 days after fertilization (Table S9). Our results showed that the expression of the *OsSR* genes was tissue-specific and development phase-dependent (Fig. 5 and Fig. S3). The genes in the SCL subfamily were mainly expressed in stems, leaves and young panicles, and showed a lower expression in grains after 5 days of pollination. Notably, the expression of *OsSC32* and *OsSC34* in SC subfamily showed leaf preferential expression, whereas the expression of *OSC25* was very low in the tested tissues. The *OsSR45a* in the SR45 subfamily was specifically higher expressed in panicles at DBF, DF, and 5 DAF, while the expression level of *OsSR45* was relatively high only in panicles at DF. As for the genes in the RS2Z subfamily, the expression of *OsRS2Z39* almost failed to be detected in both vegetative and reproductive organs. Among the remaining three genes in the RS2Z subfamily, *OsRS2Z38* was mainly expressed in the stems, leaves, and panicles before flowering, while *OsRS2Z36* was highly detected in panicles during the filling period, and *OsRS2Z37* was expressed with a relative high level in all tested tissues. The ubiquitous expression of three genes in the RSZ subfamily was observed in 8 tissues with relatively high levels, especially in stems, leaves, and grains after 10 days of fertilization. Furthermore, two RS subfamily genes were highly expressed in panicles at different developmental stages. The high expression in leaves and panicles at DBF was observed for four genes in the SR subfamily.

### Alternative splicing of OsSR genes in different tissues

It has been reported that the pre-mRNA of the SR gene which encoded the splicing regulator in different species would undergo extensive AS themselves (*Chen et al., 2019b*; *Isshiki, Tsumoto & Shimamoto, 2006*). Until now, the alternative splicing pattern of SR genes and the expression pattern of the corresponding transcripts in different tissues at different developmental stages are still poorly understood in rice. We summarized the alternative splicing of all 24 *OsSR* genes (Table S1), and the schematic diagrams of alternatively spliced transcripts of the *OsSR* genes were drawn according to the sequence information provided by the MSU-RGAP database.

To analyze the expression patterns of different transcripts produced by the alternative splicing of *OsSR* genes, we performed the RT-PCR using primers which were specific to the target genes (Table 3 and Table S7). For 11 selected *OsSR* genes, RT-PCR analysis was conducted in roots, stems, leaves, and panicles at different developmental stages. The results showed except *OsSCL25* and *OsRS2Z36*, the remaining nine genes exhibited AS actually (Fig. 6).

The *OsSCL30a* belonging to the SCL subfamily produced four transcripts, but the expression of the isoform 1 was dominant compared with other transcripts. The expression of isoform 1 could be observed in various tissues, while isoform 2, isoform 3, and isoform 4 accumulated only in the vegetative tissues including root, stem, and leaf (Fig. 6). The *OsSC34* from the SC subfamily produced three transcripts, while the expression of the isoform 1 was

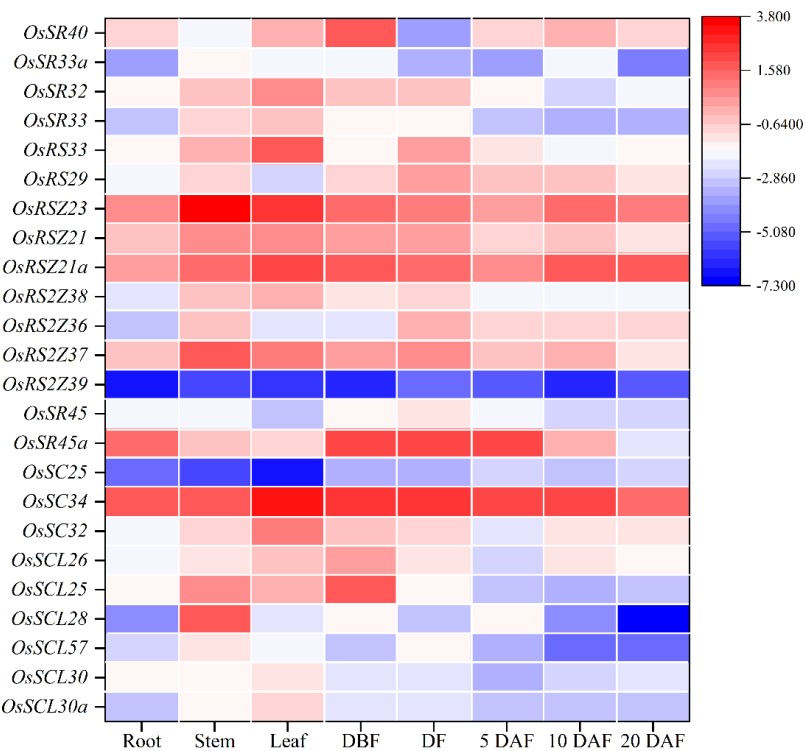

**Figure 5** **Expression profiling of the *OsSR* genes in 8 tissues based on *q*RT-PCR.** DBF, day before fertilization; DF, day of flowering; DAF, day after fertilization. *OsActin* was used as control, and each set of data contained three replicates. The comparative Δ CT values of *OsSR* genes were transformed by log2 to build the heatmap.

much more abundant than the other transcripts, which were mainly accumulated in the leaf and stem (Fig. 6). As for three genes in the RS2Z subfamily, the *OsRS2Z36* produced only one transcript (Fig. 6). The *OsRS2Z38* produced two transcripts, and the isoform 1 was predominantly accumulated in all tissues. There were four different transcripts that produced by *OsRS2Z37*, the isoform 1 or 2 was observed in all the tissues while the isoform 3 and isoform 4 were detected in different tissues except the root. Moreover, compared to the other two isoforms, the isoform 1 or 2 of *OsRS2Z37* was more abundant in all tissues. The AS pattern of *OsRSZ21* belonging to RSZ subfamily in different tissues was analyzed (Fig. 6). Two transcripts of *OsRSZ21* were observed, and the isoform 2 expressed more abundantly. The AS expression pattern of two SR genes belonging to RS subfamily in rice was detected in various tissues (Fig. 6). Compared to isoform 3 or 4, the isoform 1 or 2 generated by *OsRS29* was more abundant in all tested tissues. For *OsRS33*, a total of four different transcripts were observed in different tissues, and the isoform 1 was detected in almost all tested tissues. In the SR subfamily, *OsSR32* mainly produced isoform 1 or 2 in various tissues (Fig. 6). It was observed that the two variants generated by *OsSR33a* had equivalent expression levels within the same tissue, such as in the stem, leaf, and spikelet at 5 days after flowering, while for the other tested tissues, isoform 1 produced by *OsSR33a* was much more abundant compared to isoform 2 within the same tissue.

**Table 3  Alternative splicing pattern of 11 selected *OsSR* genes.**

| SR gene | Size of all predicted transcripts (bp) | Size of amplification product on genome (bp) |
|---|---|---|
| *OsSCL30a* | (1) 1,269 (322), (2) 2,241 (1,293), (3) 1,822 (875), (4) 2,071 (1,123) | 1,738 |
| *OsSCL25* | (1) 1,091 (491) | 1,931 |
| *OsSC34* | (1) 1,403 (695), (2) 1,416 (796), (3) 1,392 (775) | 970 |
| *OsRS2Z37* | (1) 1,332 (537), (2) 2,351 (537), (3) 2,417 (596), (4) 2,672 (851) | 1,887 |
| *OsRS2Z36* | (1) 1,312 (918) | 2,059 |
| *OsRS2Z38* | (1) 1,456 (979), (2) 1,442 (964) | 2,855 |
| *OsRSZ21* | (1) 1,398 (469), (2) 991 (583) | 957 |
| *OsRS29* | (1) 1,234 (147), (2) 1,183 (147), (3) 1,560 (571), (4) 1,579 (571) | 1,206 |
| *OsRS33* | (1) 1,349 (1,001), (2) 1422 (1,096), (3) 1,859 (1,511), (4) 1,705 (1,357) | 3,343 |
| *OsSR32* | (1) 1,042 (668), (2) 1,003 (668), (3) 1,076 (741) | 2,572 |
| *OsSR33a* | (1) 1,500 (695), (2) 1,690 (885) | 2,974 |

Notes.
The number in the parenthesis indicates the product size corresponding to the amplified primer used in this experiment.

### Expression of OsSR genes in response to abiotic stresses

According to the analysis of *cis*-acting elements in Fig. 4, the response of 24 *OsSR* genes to different environmental stress were examined (Tables S10–S14). The *OsSR* genes exhibited different expression patterns in response to the salt stress (Fig. 7A). The *OsSCL28*, *OsSC32*, *OsSR45a*, *OsRS2Z37*, and *OsRSZ21* displayed similar response patterns to salt stress. After being exposed to salt stress for 1 h to 2 days, the expression of these five genes was significantly down-regulated compared to the control. The expression levels of *OsRS2Z38*, *OsRS2Z21a*, *OsRSZ23*, *OsRS29*, *OsRS33*, and *OsSR32* were significantly induced by salt stress after 1 h of treatment. Among them, the expression levels of *OsRS2Z38* and *OsRS2Z21a* showed a steady increase relative to the control. The response of *OsSCL26* to salt stress appeared after 9 h treatment, the expression of *OsSCL26* was down-regulated dramatically. The significant and steady induction or inhibition of expression levels were not observed in other *OsSR* genes, which had the similar expression patterns to the mock treatment (Fig. S4).

Under drought stress (Fig. 7B), there was the evident increase in the expression level of *OsSCL28*, *OsSCL26*, *OsSR45a*, and *OsRSZ21a* after treatment for 1 h to 2 days. Different *OsSR* genes were responsive to drought with various degrees. The expression levels of *OsSC25*, *OsSR45*, *OsRSZ21*, and *OsRS33* were considerably up-regulated from 2 h, 9 h, 9 h, and 4 h after treatment, respectively, while *OsSCL30a* was up-regulated within 4 h of treatment, and the suppressed expression of *OsSCL30a* was observed after drought treatment for 9 h. The remaining *OsSR* genes exhibited no obvious patterns in response to drought stress (Fig. S5).

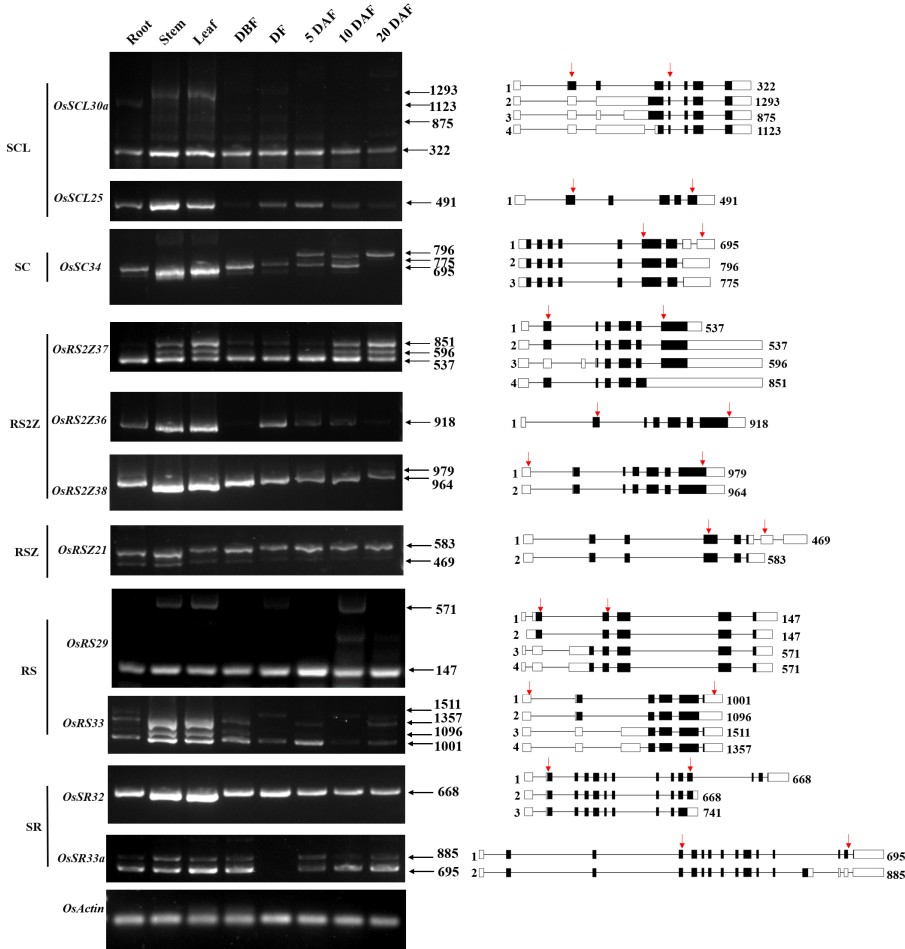

**Figure 6  Expression and AS patterns of *OsSR* genes.** DBF, day before fertilization; DF, day of flowering; DAF, day after fertilization. The numbers after the black arrows indicate the size of the amplification products. The diagrams on the right are schematic diagrams of alternatively spliced transcripts, red arrows indicate primers, the numbers on the right indicate the expected size of products. For *OsRS29*, a band with a size of about 300 bp between the target fragments of 147 bp and 571 bp was a non-specific amplification in the spikelets at 10 days after flowering according to the sequencing result.

The expression of *OsSC25*, *OsSR45a*, *OsRSZ21a*, and *OsSR33* were induced by the cold treatment (Fig. 8A). The induced expression of *OsSC25* peaked at 6 h after treatment. The induction of *OsSR45a* was strong, the expression level of *OsSR45a* increased by more than 10 times compared to the control within 1 h to 9 h after treatment. Under cold stress, the expression levels of *OsSCL30*, *OsSCL28*, *OsSCL26*, and *OsRSZ23* were remarkably decreased. The *OsRSZ23* showed an exaggerated response to the cold treatment, its expression was almost completely suppressed under low temperature. Furthermore, the expression levels of other *OsSR* genes fluctuated, but the changes were slight between treatment and control (Fig. S6).

The *OsSR* genes were responsive to high temperature with various patterns and degrees (Figs. 8B and S7). Heat treatment induced the significant down-regulation of *OsSCL30a*,

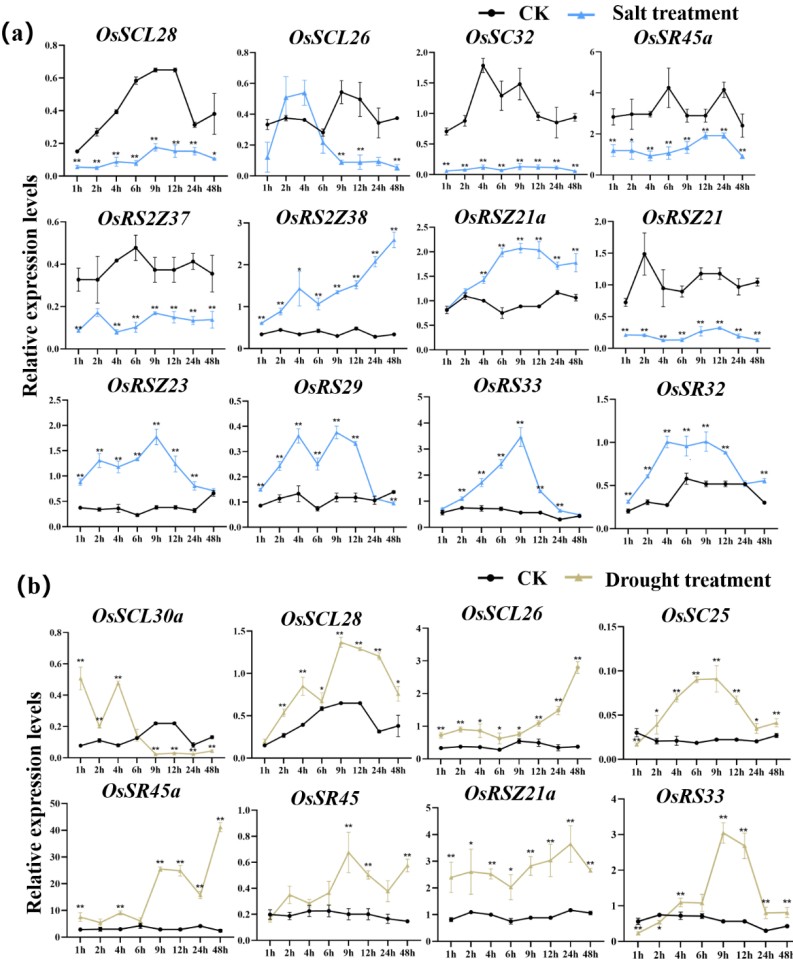

**Figure 7** **Expression of *OsSR* genes in response to salt (A) and drought (B) stress.** *OsActin* was used as control. Error bars represent mean ± SE of three biological replicates. *$P < 0.05$ and **$P < 0.01$ indicate significant differences compared with CK determined by Student's $t$-test.

*OsSCL30*, *OsSCL26*, *OsRS2Z37*, *OsRSZ23* and *OsRS33* (Fig. 8B). Notably, the expression of *OsSCL26* gene was almost completely suppressed under heat stress. The results showed that heat treatment significantly upregulated the expression of *OsSCL25* and *OsSC32* for 1 h to 2 days (Fig. 8B). After exposure to heat stress within 6 h, the expression of *OsSR45a* was remarkably induced, and it was observed to be down-regulated after 9 h of treatment. The expression of *OsRS29* was up-regulated within 1 day of heat treatment and began to decrease after 1 day. The response of *OsRSZ21* and *OsSR33* to heat stress appeared after 9 h of treatment, showing a significant down-regulation (Fig. 8B).

### Expression of OsSR genes in response to hormones

The expression patterns of *OsSR* genes under different phytohormone treatments were investigated by *q*RT-PCR (Figs. 9, S8, S9, Tables S10 and S15–S16). We focused on two hormones, ABA and GA, which are essential for plant growth and development. The results showed that *OsSCL25*, *OsRSZ21* and *OsSR33a* were significantly induced by GA.

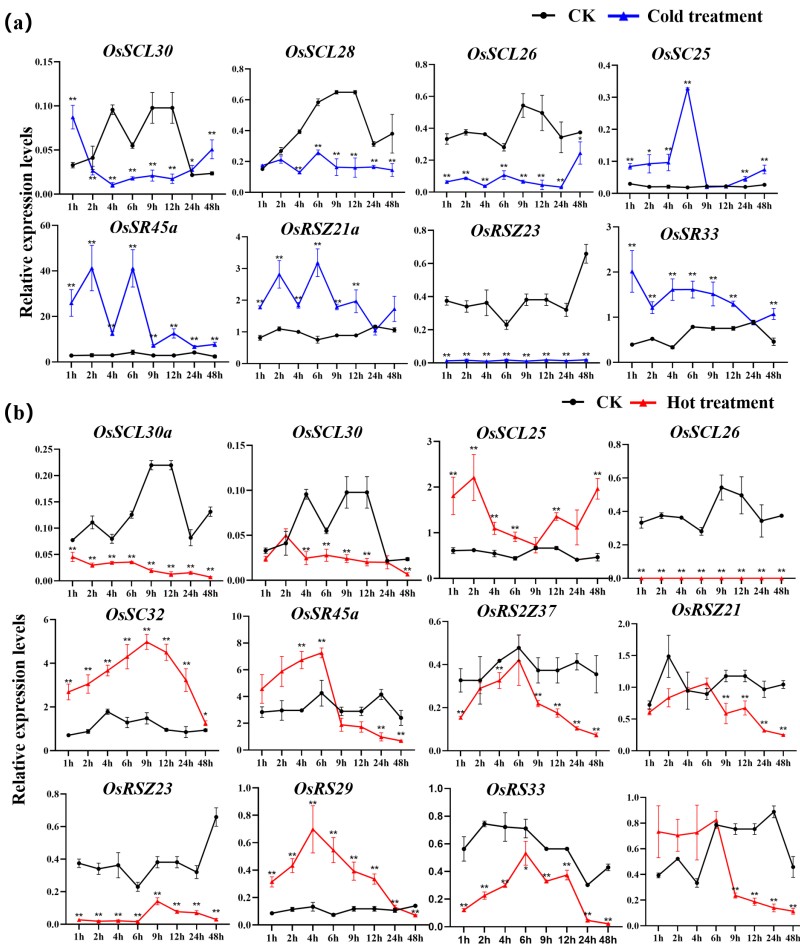

**Figure 8 Expression of *OsSR* genes in response to cold (A) and hot (B) stress.** *OsActin* was used as control. Error bars represent mean ± SE of three biological replicates, *$P < 0.05$ and **$P < 0.01$ indicate significant differences compared with CK determined by Student's $t$-test.

*OsSCL28*, *OsSR45a*, *OsSR32* and *OsSR33* were induced by GA after treatment for 1 h to 12 h (Fig. 9A), the induced peak values appeared at about 9 h. After being treated with GA for approximately 5 h, there was a considerable decrease in the expression level of *OsRS2Z38*, then the level gradually increased from 24 h time point and recovered to the similar level compared to the control at 48 h time point (Fig. 9A).

Only a few *OsSR* genes showed obvious response patterns under exogenous ABA treatment. As compared with the expression under mock treatment, the ABA treatment resulted in a significant increase in that of *OsSCL25* and *OsSR45a*, while the change in expression of *OsRS2Z36* was found to be inverse, which was shown to be suppressed by ABA treatment (Fig. 9B).

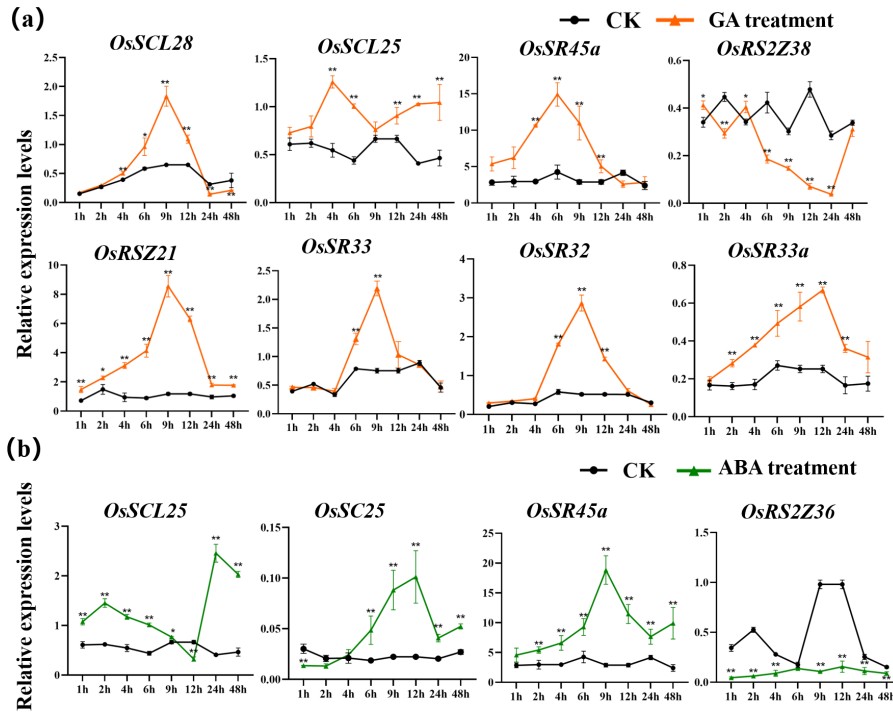

**Figure 9 Expression of *OsSR* genes in response to hormones.** *OsActin* was used as control. Error bars represent mean ± SE of three biological replicates. *$P < 0.05$ and **$P < 0.01$ indicate significant differences compared with CK determined by Student's *t*-test.

## DISCUSSION

SR proteins, which work as the splicing regulators, play indispensable role in constitutive and alternative splicing of pre-mRNA. The SR gene family has been identified in many plant species, such as *Arabidopsis*, rice, wheat, maize, cotton and longan (*Chen et al., 2019b*; *Chen et al., 2020b*; *Jin, 2022*; *Wei et al., 2022*). In this study, we focused on the 24 SR genes in rice, including their classification, gene and protein structure, chromosomal location, evolution, *cis*-elements, expression profiles, and response to abiotic stresses and hormones.

### Domains and physicochemical properties of OsSR proteins
The SR proteins were evolutionarily conserved at the structural level. Previous studies have shown that these conserved domains are essential for the protein to function properly in plants. In *Arabidopsis*, N- or C-RS domains were necessary for the accurate nuclear localization of atSR30 and atSR45a (*Mori et al., 2012*). It has been reported that RSZp22, which was the member of the RSZ subfamily in *Arabidopsis*, displayed speckle-like distribution and localization in the nucleus, and the realization of this accurate localization was inseparable from the presence of RRM and zinc-knuckle in the protein sequence of RSZp22 (*Rausin et al., 2010*). The analysis of domain and protein structural characteristics in this study laid the foundation for further understanding the function of OsSR. *OsSR* genes with closer evolutionary relationship were similar in protein domain distribution and protein structure, indicating that *OsSR* genes belonging to the same subfamily would have

similar functions. We found that the OsSR proteins in the same subfamily showed similar motif arrangements and considerable variation among different subfamilies (Fig. 1B). All OsSR proteins were found to have the RRM domain, and the Zn_C$_2$HC domain was contained in the RSZ and RS2Z subfamilies (Fig. 1). In addition, the distribution of domains in OsSR proteins were significantly different among different subfamilies, which may be related to the functional differentiation of OsSR proteins. However, there are few studies on the effect of conservative domains on the proper function of OsSR proteins. Further research could be conducted to edit the RRM, SR or Zn_C2HC domain of *OsSR* genes by gene editing technology.

The SR proteins are essential nuclear localized proteins that function as splicing factors in splicing of precursor mRNA (*Misteli, Cáceres & Spector, 1997*). In the current study, subcellular localization of most OsSR proteins was in the nucleus (Table 1), indicating the OsSR proteins could function as splicing factors as those in other species. Intriguingly, SR proteins were also involved in post-splicing activities, which were achieved through continuous shuttling between the nucleus and the cytoplasm (*Huang & Steitz, 2001*; *Michlewski, Sanford & Cáceres, 2008*; *Swartz et al., 2007*). The state of phosphorylation and dephosphorylation was the key factor affecting the dynamic subcellular localization of SR proteins (*Jin, 2022*; *Mori et al., 2012*). For example, studies have proved that RSZp22 in *Arabidopsis thaliana* was a nucleocyto-plasmic shuttling protein, and its shuttling property has been analyzed in details (*Rausin et al., 2010*; *Tillemans et al., 2006*). In our study, the OsSR proteins contained different amounts of phosphorylation sites (Table 1), indicating OsSR proteins could function by shuttling between nucleus and the cytoplasm. Further research is needed on dynamic distribution of OsSR proteins and how this process affects post-splicing activities including mRNA export.

## The orthologous SR gene pairs between rice and other species provide insights into the evolution and function of the *OsSR* genes

Synteny refers to the distribution or arrangement of homologous genes within one specie or among different species (*McCouch, 2001*). Collinearity analysis in this study revealed the distribution of orthologous genes of SR genes between rice and other species, which helped us further understand the origin of the *OsSR* genes. Compared to a larger number of orthologous SR gene pairs identified between rice and other monocotyledonous plants, no ortholog was found in soybean and only four orthologs of *OsSR* genes were found in *Arabidopsis* (Fig. S2, Fig. 3 and Table S4), suggesting that the development of orthologous SR pairs was more probable to occur after the divergence of dicots and monocots. Evidently, multiple *TaSR* genes were identified as orthologs of single *OsSR* gene. For instance, *TaSR4D*, *TaSR7A*, *TaSR6B*, *TaSR7D* and *TaSR14D* were the orthologs of *OsSR33* (Table S4), indicating the expansion of *OsSR* genes may occur before that of wheat. These orthologous SR genes in different species may have similar functions that are involved in constitutive and alternative pre-mRNA splicing, and post-splicing activities.

The orthologous genes in different species may have originated from a common ancestor. The sequences of orthologous genes are conserved, indicating the conservation of function of these genes (*Tang et al., 2008*). Understanding the function of these orthologous genes
is helpful to reveal and explore the function of *OsSR* genes. Among the orthologous SR genes identified in other species in this study, there have been some reports on their gene functions. *AtSR34* in *Arabidopsis*, the orthologous gene of rice *OsSR33a* and *OsSR33* has been reported to be related to heat stress response (*Ling et al., 2018*).

## AS-NAGNAG events were not frequent on *OsSR* genes

NAGNAG splicing produces two distinct isoforms that are distinguished by three nucleotides (NAG, N = A, C, G, T). Due to the fact that SR genes would undergo alternative splicing, one SR gene can generate several transcripts that encode different isoforms, which give the spliceosome greater spatial flexibility, and influence the outcome of splicing (*Graveley, 2000*). Protein diversity induced by the AS-NAGNAG contributes to this flexibility to some extent. The previous studies have reported that NAGNAG acceptor motifs were frequent in human genes and SR genes in *Arabidopsis* (*Hiller et al., 2004*; *Schindler et al., 2008*). Here, we screened for NAGNAG acceptor tandems in *OsSR* genes. A total of 19 NAGNAG acceptors were identified in 14 *OsSR* genes, belonging to seven subfamilies (Table 2). However, AS-NAGNAG events were only observed at the location of the three acceptors. But we only summarized the AS-NAGNAG events of *OsSR* genes under normal growth conditions. Notably, the different tissues and the change of environmental conditions could affect the alternative splicing rate occurring at the NAGNAG acceptor. For example, the AS-NAGNAG events in *Arabidopsis* may be mediated by the organ and condition-specific differences of the spliceosome (*Schindler et al., 2008*). Thus, how these factors affect NAGNAG alternative splicing in *OsSR* genes remains to be evaluated.

## The AS pattern of *OsSR* genes varied with different tissues

Most SR genes themselves undergo extensive alternative splicing (*Reddy & Shad Ali, 2011*). This study investigated the alternative splicing pattern as well as the expression of different transcripts produced by the *OsSR* genes in both vegetative and reproductive tissues (Fig. 6). The alternative splicing patterns of *OsSR* genes were tissue-specific, which means the expression levels of different transcripts produced by the same *OsSR* genes varied greatly in different tissues, but most *OsSR* genes mainly express one transcript in each tissue. Previous studies performed the investigation of all splicing variants of SR genes in *Arabidopsis*. Notably, the majority of these alternative splicing occurred within the coding region of SR genes, and the AS type on SR genes in *Arabidopsis* was mainly intron retention (*Palusa, Ali & Reddy, 2007*). Interestingly, we found in most cases, AS events of *OsSR* genes occurred in the 3′ or 5′ untranslated regions, which would not cause the corresponding genes to generate new protein coding sequences (Fig. 6 and Table S1). We speculated that such splicing may have an impact on the expression and stability of precursor mRNA (*Jin, 2022*). Nevertheless, some *OsSR* genes such as *OsSCL30a*, *OsRS2Z37*, *OsRS2Z38*, *etc.*, could undergo alternative splicing in the coding region and generate transcripts with different CDS, which means they could encode different proteins (Fig. 6 and Table S1). Moreover, these different transcripts produced by the same *OsSR* gene had tissue expression specificity, and some transcripts could only be detected in specific tissues. For example, isoform 3 and isoform 4 produced by *OsRS29* were only detected in stem, leaf and spikelets at 10

days after flowering (Fig. 6), indicating that the proteins encoded by these transcripts were only expressed in specific tissues. Studies have shown that different transcripts of one gene produced by alternative splicing may perform distinct functions. The SR gene *SR45* in *Arabidopsis* could produce two transcripts, and *SR45.1* played a role in flower development, while *SR45.2* was involved in regulating root growth and development (*Zhang & Mount, 2009*). Whether there are functional differences between different transcripts produced by the same *OsSR* gene in rice needs further exploration and research.

### *OsSR* genes may function in plant growth, response to hormones and abiotic stresses

In the current study, we found that the majority of *OsSR* genes expressed extensively with different levels in stems, leaves, or spikelets. The expression patterns of different *OsSR* genes were tissue and development stage dependent, indicating their specific functions. Based on the detection results of gene tissue-specific expression (Fig. 5), we speculated that SCL, SC, and RS2Z subfamily genes may be involved in regulating the development of vegetative organs in rice, while the *OsSR* genes in SR45, RSZ, and RS subfamily were more likely to participate in regulating the formation and filling of grains in rice.

The growth of plants could be profoundly influenced by a variety of environmental conditions. The transcription levels of related genes could be induced, repressed or regulated by various stresses (*Palusa, Ali & Reddy, 2007*). However, the expression profiles of *OsSR* genes under various stresses have not been detected till now. Inducing or inhibiting the binding of transcription factors to the corresponding *cis*-acting sites in the gene promoter region to regulate the expression of downstream genes is an important mechanism to respond to environmental changes (*Riechmann et al., 2000*). The identification of *cis*-elements provides clues for determining gene expression patterns under different kinds of stresses. The promoter analysis in this study suggested that *OsSR* genes played important roles in various stress responses in rice. In the promoter region of *OsSR* genes, different types of *cis*-acting elements were discovered, including 10 hormone-responsive and 5 stress-responsive elements (Fig. 4 and Table S5). The expression of genes is influenced by plant growth stage and environment. For the gene whose function is unknown, the distribution of *cis*-elements in the genes' promoter region could not directly reflect gene expression, but could provide information for us to further explore the pathway of this gene participating in response. Our results showed the expression of some of *OsSR* genes were changed after abiotic or hormone treatment (Figs. 7, 8 and 9), indicating that they modulated the response to stresses in rice. These results lay a foundation for further understanding the function of *OsSR* genes. For example, our experiment showed that under salt treatment, the expression of *OsRS33* was significantly upregulated (Fig. 7), which was consistent with the previous study that *OsRS33* gene knockout lines were more sensitive to salt stress compared with the wild type (*Butt et al., 2022*). We found *OsSCL30* was obviously and continuously suppressed by the cold treatment, and in fact, *OsSCL30* was related to cold tolerance in rice, overexpression of *OsSCL30* reduced the tolerance of rice seedlings to low temperature (*Zhang et al., 2022*).

Besides, we observed that some *OsSR* genes respond to a variety of stresses simultaneously. A summary of differentially expressed *OsSR* genes in response to various abiotic stresses was provided in the Table S8. The results indicated that response patterns to abiotic stresses of *OsSR* genes were time-dependent and varied among different genes. *OsSCL30*, *OsSCL26* and *OsRSZ23* responded to both cold and heat stress, suggesting that the expression of these genes was affected by ambient temperature (Fig. 8, Table S8). Moreover, the expression of *OsSC25* and *OsRSZ21a* genes was affected by both drought and cold stress, while the expression of *OsSC32* and *OsRSZ23* was affected by both salt and temperature stress (Figs. 7 and 8 and Table S8). In addition, *OsSCL25* and *OsSR45a* were found to respond to GA and ABA simultaneously (Fig. 9, Table S8). In *Arabidopsis*, *SR45a* responded to ABA and abiotic stresses (*Cruz et al., 2014*; *Ling, Mahfouz & Zhou, 2021*). Consistently, we found that *OsSR45a*, a member in SR45 subfamily, could also respond to multiple stresses simultaneously. The results showed that ABA, GA, salt, drought and temperature stress significantly affected the expression level of *OsSR45a* (Figs. 7, 8 and 9), indicating that expression pattern of *OsSR45a* were stress-dependent. Altogether, these results strongly suggest that *OsSR* genes are critical in response to environmental signals in rice, and the function and mechanism of *OsSR* genes could be further studied based on the results in this study.

## CONCLUSIONS

In this study, the comprehensive analysis on *OsSR* genes gave some insights on their characteristics and functions. It showed that 24 *OsSR* genes were distributed in seven different subfamilies based on the phylogenetic analysis. Gene structures of *OsSR* genes, distribution of domains, and protein structure of OsSRs were conserved within each subfamily. There were six segmental duplicated *OsSR* gene pairs (50%) in the rice genome, indicating segmental duplication played an overwhelming role in the expansion of *SR* gene family in rice. Most of *OsSR* genes would undergo AS and the AS patterns varied with different tissues. The majority *OsSR* genes were expressed in different tissues, while their expression levels varied substantially among different organs, suggesting their extensive functions in vegetative growth or spikelet development in rice. Furthermore, the expression patterns of *OsSR* genes would change significantly under abiotic stress or hormone treatment, indicating that *OsSR* genes may participate in the hormone/abiotic stress signaling pathway in rice. The current results will be helpful for better understanding and further study of *OsSR* genes.

### Funding

This study was supported by the Science and Technology Office of Zhejiang Province, China (Project no. 2021C02063-6). The funders had no role in study design, data collection and analysis, decision to publish, or preparation of the manuscript.

## Grant Disclosures

The following grant information was disclosed by the authors:
Science and Technology Office of Zhejiang Province, China: 2021C02063-6.

## Competing Interests

The authors declare there are no competing interests.

## Author Contributions

- Rui Gao conceived and designed the experiments, performed the experiments, analyzed the data, prepared figures and/or tables, and approved the final draft.
- Yingying Lu performed the experiments, prepared figures and/or tables, and approved the final draft.
- Nan Wu performed the experiments, prepared figures and/or tables, and approved the final draft.
- Hui Liu performed the experiments, prepared figures and/or tables, and approved the final draft.
- Xiaoli Jin conceived and designed the experiments, authored or reviewed drafts of the article, and approved the final draft.

## Data Availability

The raw data is available in the Supplementary Files.

## Supplemental Information

Supplemental information for this article can be found online at http://dx.doi.org/10.7717/peerj.16193#supplemental-information.

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
