# Peer review of "Comprehensive study of serine/arginine-rich (SR) gene family in rice: characterization, evolution and expression analysis"

_PeerJ, doi:10.7717/peerj.16193_

## Round 0.1 · original submission · Minor Revisions

Dear Authors,

Your work is a very good example of Omics data reuse and data mining in combination of the experimental work. I invite you to revise your manuscript and address all the comments from the three reviewers. In addition, here are a few editor's comments:

1. I am surprised that authors have retrieved the SR family genes from RAP DB, but in the Table 1 they have listed LOC_geneID of the MSU project. Please provide RAP Gene ID of all the 24 OsSR genes in Table 1 in addition to LOC_gene IDs. The LOC gene IDs of the rice genes are derived from the MSU project that has not been updated since 2012. The RAP DB (https://rapdb.dna.affrc.go.jp/index.html) is being updated routinely and has revised gene annotation and gene model updates. The rice community, UniProt and Gramene database all depend on the updated RAPDB gene annotations.

2. Also address if there is difference in the gene models between RAP DB and MSU-LOC gene models and if this affects your alternative splicing analysis.

3. The subcellular location of the proteins using ExPASy Protparam is all predictive. I suggest that you look into the literature, if subcellular location of any protein has been experimentally verified and if there is any proteome data that supports the subcellular location of any protein at https://croppal.org/ . In Table 1 highlight/bold the experimentally verified subcellular location.

4. How well does promoter cis-elements and gene expression data match?

5. Based on your study, could you summarize a gene summary/description for all the genes, and if any of this information can be used for improving the gene annotation of the SR gene family members or their association with any pathways or molecular process. This information can be submitted to UniProt or RAP-DB or Gramene database. For an example see a previous study published in the PeerJ : https://peerj.com/articles/11052/ .

·

Basic reporting

The authors have performed a comprehensive characterization and expression analysis of the SR gene family in rice. The results of this study are important for future functional validation studies on these SR proteins and is a useful resource that can be used to guide future studies on SR gene characterization in other plants. After the recommended revisions are made, this manuscript can be accepted for publication.



The manuscript needs to be revised to improve the sentence structure and grammar throughout. The following points should be noted to improve the grammar throughout the manuscript:

Line 38: the meaning of AS should be described at its first appearance in line 36, and not in line 38

Line 70-71: needs restructuring

Line 83: “expression of...”

Line 85: don’t start sentence with ‘and’; this sentence is also too long and should be broken into two

Line 97: needs restructuring

Line 103: ‘functions’ should be ‘functional’

Line 130: “relationship of between”

Line 154: “... to an incubator...”

Line 162-163: don’t start sentence with ‘and’

Line 175: remove this sentence. It sounds as though the SR genes were not identified until now, but it was already identified in 2010; this study seeks to characterize them.

Line 185: needs restructuring

Line 209-211: needs restructurings

Line 214: ‘terminus’ and ‘subfamily’ should be plural

Line 229: don’t start sentence with ‘and’

Line 260: don’t start sentence with ‘and’

Line 265: ‘... that are involved...”

Line 270: “genome” should be plural

Line 283: ‘response’ should be ‘respond’

Line 285: “...types of...”

Line 294: “... that respond...”

Line 305: ‘respond’ should be ‘responding’

Lines 308-309: needs rewording

Line 313: should be ’tissue-specific’

Lines 324-325: needs rewording

Line 327: “... genes were highly expressed...”

Line 328: don’t start sentence with ‘and’

Figure 2 legend: “Chr1- Chr24” should this read Chr12 or Chr24?

Line 335 and 342: “...developmental stages...”

Line 357, 360: abundant should be abundantly

Line 369: ‘suggesting’ should be ‘suggests’

The abbreviation ‘CK’ is used in the legend for Figure 8 and in several other tables/ figures. The meaning of this abbreviation cannot be found in the manuscript.

Lines 375 and 378: needs rewording

Line 425: needs rewording

Line 433; “... protein to...”

Line 443: ‘subfamily’ should be ‘subfamilies’

Line 444; ‘properly’ should be ‘proper’

Line 446: ‘analysis of...”

Line 474: “may have originated”. This sentence is also too long.

Line 495: needs rewording

Line 502: ‘study’ should be plural

Line 514: “are only”

Line 521: “function in”

Line 522: needs rewording

Line 560: should be ‘characteristics’

Line 569: needs rewording

Experimental design

The method is described with sufficient detail and the experiments were suitable for answering the research question. Was the semi-quantitative RT-PCR conducted with replicates? If so, this could be stated in the method.

Validity of the findings

no comment

Additional comments

no comment

·

Basic reporting

Review

1. The writing was clear. The areas needing work have been marked in the attached .pdf.
2. The Intro and background were relevant and well-referenced.

Experimental design

3. The structure conforms to the PeerJ standards.
4. The figures are detailed and meaningful.
5. The primary research is within the scope of the journal.
6. The research question is very relevant and meaningful.
7. The investigation is rigorous, and the methods provided in enough detail to replicate the research.
8. I commend the authors on a well-reasoned and complete study.

Validity of the findings

Appropriate replicates performed.

Additional comments

Points of concern are identified within the . pdf
1. Line 18 in the abstract.
2. Of the keywords, the word ‘stress’ is ambiguous. Authors should consider ‘abiotic stress’.
3. Lines to check within the Intro are, 41, 42,71,76,93,96, 97,99,100,101,102, 103,
4. Lines in Materials and Methods: 108, 113,130, 131,140-143, 149, 152,153, 155, 162,165,167
5. Lines in Results: 183, 184, 186,197,198, 218-220, 230, 236, 237, 245, 262, 272, 273, 282-286, 297,334, 357, 360, 375,
6. Lines in Discussion: 425,436, 444, 447, 450, 468, 480, 484-91,494, 514, 563, 566,
I did not verify that the references are all in the required format.

·

Basic reporting

Satisfactory.

Experimental design

Satisfactory.

Validity of the findings

Satisfactory.

Additional comments

The manuscript by Gao et al. described the comprehensive study of Serine/Arginine-Rich (SR) gene family in rice. The study is well-designed and well-presented. Although the genome sequence of rice is available since a while, still a genome-wide analysis of key gene families are still important. I have a few comments for the betterment of the article.

1. Abstract can be shortened. Please retain only key and significant findings here.

2. In method section, please elaborate a bit more on the extraction and selection of sequences (line 107).

3. The phylogenetic analysis has been done for the rice SR genes only. I would advise the authors to include other related plant SR genes, such as maize or wheat.

---

## Round 0.2 · Minor Revisions

Dear Authors,

Please address all comments and provide clarification for the comments made by the reviewers. Please include RAP gene ID for all the rice genes included in this manuscript. If you are using MSU gene IDs (that Start with LOC_xxx), then also provide RAP geneIDs. If there is any discrepancy in the gene model between the two gene nomenclature system, then address those as well.

·

Basic reporting

Review 2023:04:84270:1:1:NEW 2 Jul 2023
Section 2.3 – Check all units. Some are spaced from the quantities some are not. For example, lines 148 -150. Such errors are present throughout and should be double checked for consistency.
Section 3.1 – line 186, delete ‘except for’ and ‘which’.
Based on Table 1, OsSR45 is expressed in the nucleus. Change line 187 by inserting ‘ and nucleus’. The sentence should read ‘…OsSR45 was …..in the chloroplast and nucleus. The remaining… localized ‘only’ in ‘the’ nucleus.

Line 194- since there are two OsRS33, the RAP_locus id should be added.
OsRS33 (LOC_Os02g03040.1) and OsRS33 (LOC_Os0s07g47630.1)
Line 196- It would be valuable to report that all of the OSR proteins were phosphorylated with numbers of phosphorylation sites ranging…..(Table 1).
Lines 198- 200 starting with ‘The results inferred….are not results but discussion’. Recommend deleting.
Line 231 – delete ‘for example’ and end the sentence after ‘structures’. Delete n’indicating…..function’.
Lines 233 – 236 ending in ‘OsSR genes’ fits better in Discussions.
Line 237- ‘analyzed using GSDS online tool’ does not belong in Results. It can be deleted and the sentence presented as ‘The genetic structure of the 24 OsSR genes differed in nucleotide sequence…’
Line 240- has a mistake. OsSR33a has more introns and exons than OsRSZ23. The sentence reverses this information.
Lines 239 – 245 are confusing
Lines 246 -249 are Discussion rather than Results.
Lines 251-252 ending with …2004) can be moved to Discussions
Line 254- delete switch ‘were’ to ‘are’.
Line 255 – Transfer ‘Totally ….2002)’ to discussions.
Lines 259-260 Keep only ‘ The Ka/Ks ratios…..Table S2.’ In Results. The rest belongs in Discussions
Section 3.4: Keep ‘The synthenic …….(Fig 3). And ‘In addition… Table S30’ in this section. Transfer everything else to the Discussion section .
Section 3.5: Start at line 282 end at end of 289. Move all of lines 274 to 281 to Discussions
Section 3.6 Keep paragraph starting with line 295 ‘The analysis…to line 315 ending ‘promoters’. All else ‘Based on these findings…’ can be transferred to Discussions.
Section 3.7 Line 319- end at genes. Transfer from ‘suggesting …response to plants’ to discussions.
Section 3.7.1 Line 329- missing verb could use ‘were’ before mainly.
Lines 332- 334: This reviewer disagrees with the statement on those lines. According to Fig 5., OsSR45 is not similarly expressed to OsSR45a. The statement made in the text applies only to OsSR45a.
Line 334- delete ‘was’. Can delete ‘ indicating ….luxury gene’. If using ‘luxury gene’ need to define at some point.
Line 336- OsRS2Z36 did not match the Z37 and Z38 in that Z36 is not expressed in leaf as stated in the text.
Section 3.7.2
Please review data
The results that OsRS2Z37 produces 4 different transcripts (line 363 and 364) is not supported either by Fig 6 or Table S6. There appears to be only three different transcripts based on the presence of only three amplicons. This reviewer assumes that the transcripts in Fig 6 are labeled as isoforms based on the amplicon sizes shown in the gel. There are two transcripts labeled 537. How did the authors distinguish the transcripts for these amplicons as 1soform 1 and 2? The red arrows on the transcripts indicates only one amplicon for both transcripts for those two primers.
OsRS2Z38 is problematic to decipher. The arrows in Fig 6 do not appear to be properly positioned. It is unclear whether the amplicon in stem is 964 as it points to a higher band in 20 DAF (amplicon 979) than the band in stem and leaf. The statement line 363- ‘ isoform 2 was predominantly accumulated in all tissues ‘ is not supported by the data. Since isoform 2 (amplicon 964) is only in stem and leaf.
Line 367: For RSZ21 there are two transcripts isoform 1 (amplicon 469) and isoform 2 (amplicon 583). The data does not support the statement ‘line 368-9 ‘ and the isoform 1 expressed more abundantly’.
Line 370 -1identifies two transcripts (isoform 1 and isoform 2) for RS29 that ‘were more abundant in all tested tissues. The data supports only isoform 1 This is a problematic statement, since the amplicon generated by the primers used on both transcripts would produce the same size of 147. Given this design, this reviewer is of the opinion that the authors are not positioned to discuss variations in expression of alternatively spliced transcripts that generated the same size amplicon.
Additionally, this reviewer observes at least 3 other amplicons in the 10 DAF sample. There appears to be two other faint ones. However, it is not clear why the authors have ignored the third one in the mid region above 147.
Line 372: The statement that the 4 transcripts of OsRS33 ‘could be detected in different tissues except the root’ is not supported by the data in Fig 6. The majority of alternative transcripts are missing from DBF, DF, 5DAF, and 10 DAF.
Line 373: The statement ‘ OsSR32 mainly produced isoform 1 and 2 in various tissues’ is not supported by the amplicon data since both isoform 1 and 2 are listed as identified with amplicon 668. To make such a statement the experimental design should have included primers that could distinguish between iosforms 1 and 2. Maybe there is a smaller amplicon in the stem and leaf samples. However, it is not labeled. It is unclear in Table S7 where the predicted sizes of two proteins were obtained.
Line 374-6: This reviewer does not accept the statement of the results for OsR33a that the two variants ‘had equivalent expression ….. in other tissues. As shown in Fig 6, amplicon 885 is differentially expressed in root, DF, 10 DAF, and 20 DAF.
Lines 377-379 belong in the Discussion section.
Section 3.7.3
Line 381 should be removed to Discussion. Also the data for this section is displayed in Fig 7 and Fig S4, not Fig 4 as stated in line 381.
Line 388 description of ‘steady increase relative to the control’ is not supported by the data. The description of expression for OsSCL26 (line 390) as ‘appeared after 9 hr ….is not supported by the data. The data Fig. 7a shows expression of OsSCL26 rising or appearing above CK at about 2hr.
Lines 419-420 should be moved to Discussions
Section 3.7.4.
Lines 422-425 can be moved to Discussions
Section 4 Discussion
Line 442 ‘abiotic stresses and hormones’
Line 515 ‘and generate’ not generated
Line 577 add ‘were’ expressed; add ‘s’ to level

Experimental design

no coment

Validity of the findings

in section 1

Additional comments

The results section needs to be carefully reviewed. The data is present, but the reporting of the results is problematic

·

Basic reporting

Satisfactory

Experimental design

Satisfactory

Validity of the findings

Satisfactory

Additional comments

Authors have addressed my comments in this revised version of the manuscript satisfactorily.

---

## Round 0.3 · accepted · Accept

The authors have addressed all the comments made by the reviewers and revised the manuscript accordingly.

·

Basic reporting

Problems addressed

Experimental design

accepted

Validity of the findings

accepted

Additional comments

line 411 delete actually
line 421 delete that